# Elevated Hepcidin Expression in Human Carotid Atheroma: Sex-Specific Differences and Associations with Plaque Vulnerability

**DOI:** 10.3390/ijms25031706

**Published:** 2024-01-30

**Authors:** Xi-Ming Yuan, Nargis Sultana, Moumita Ghosh-Laskar, Wei Li

**Affiliations:** 1Occupational and Environmental Medicine, Department of Health, Medicine and Caring Sciences, Linköping University, 581 85 Linköping, Sweden; ximing.yuan@liu.se; 2Laboratory Medicine, Linköping University Hospital, 581 85 Linköping, Sweden; nargis.sultana@regionostergotland.se; 3Research & Development, Cytiva, 753 23 Uppsala, Sweden; moudocs@gmail.com; 4Obstetrics and Gynaecology in Linköping, Department of Biomedical and Clinical Sciences, Linköping University, 581 85 Linköping, Sweden

**Keywords:** atherosclerosis, apoapsis, hepcidin, macrophages, male, oxidative stress, plaque instability, sex difference

## Abstract

Hepcidin is upregulated by increased body iron stores and inflammatory cytokines. It is associated with cardiovascular events, arterial stiffness, and increased iron accumulation in human atheroma with hemorrhage. However, it is unknown whether the expression of hepcidin in human carotid plaques is related to plaque severity and whether hepcidin expression differs between men and women. Carotid samples from 58 patients (38 males and 20 females) were immunostained with hepcidin, macrophages, ferritin, and transferrin receptor. Immunocytochemistry of hepcidin was performed on THP-1 macrophages exposed to iron or 7betahydroxycholesterol. Hepcidin expression significantly increases with the progression of human atherosclerotic plaques. Plaques of male patients have significantly higher levels of hepcidin. Expressions of hepcidin are significantly correlated with the accumulation of CD68-positive macrophages and transferrin receptor 1 (TfR1) and apoptosis. In vitro, hepcidin is significantly increased in macrophages exposed to iron and moderately increased following 7-oxysterol treatment. In the cultured cells, suppression of hepcidin protected against macrophage cell death, lysosomal membrane permeabilization, and oxidative stress. Hepcidin may play a crucial role in the development and progression of atherosclerosis. The differential expression of hepcidin in male and female patients and its significant correlations with plaque severity, highlight the potential of hepcidin as a biomarker for risk stratification and therapeutic targeting in atherosclerosis.

## 1. Introduction

Iron is essential for physiological processes, but it also plays a key role in numerous pathological conditions, including cardiovascular diseases. Dr. Sullivan proposed the iron hypothesis in 1981 and suggested that iron deficiency may play a protective role against heart disease in women [1]. Despite conflicting epidemiological studies, experimental investigations have shown the accumulation of iron in atherosclerotic lesions [2], which, in turn, may lead to iron-driven oxidative stress, activation of multiple signaling pathways in atherosclerotic vessel walls, and cause endothelial dysfunction and atherosclerosis [3].

Hepcidin, a peptide with antimicrobial properties, has been recognized as a key regulator of iron metabolism in the body [4,5,6]. Hepcidin levels are upregulated in response to increased body iron stores and inflammatory cytokines and have been reported to be associated with cardiovascular events and arterial stiffness in chronic hemodialysis patients [7]. Additionally, serum hepcidin levels are elevated in individuals with metabolic syndrome [8] and are associated with the presence of plaque in postmenopausal women [9]. In a mouse model, hepcidin was found to be upregulated in carotid plaques and acted as a positive regulator of atherosclerotic plaque instability [10]. Studies have shown that in cultured macrophages, oxidized low-density lipoprotein (ox-LDL) can increase reactive oxygen species formation, inflammatory cytokine production, apoptosis, and upregulate hepcidin expression [10]. Furthermore, increased hepcidin expression has been observed in human carotid plaques with hemorrhagic characteristics and associated iron accumulation [11,12]. Despite the widely acknowledged variances in the occurrence and complexities of atherosclerosis between males and females, there exists a scarcity of evidence within clinical and preclinical research to examine the underlying mechanisms that categorize sex as a biological determinant in atherosclerosis [13]. Currently, it is unidentified whether hepcidin expression in human carotid plaques is related to plaque severity and whether there are any sex-specific differences in hepcidin expression. Additionally, it remains unknown whether hepcidin expression in macrophages is influenced by oxysterols that are relevant to atheroma formation.

The present study aims to investigate several factors related to hepcidin expression in human carotid plaques. Specifically, the study will explore (1) the potential relationship between hepcidin levels and plaque severity, (2) whether there are any sex-specific differences in hepcidin expression within carotid plaques from men and women, and (3) the potential association between hepcidin expression and macrophage apoptosis induced by 7betahydroxycholesterol and the underlying mechanisms by focusing on lysosomal membrane permeabilization (LMP) and oxidative stress. By investigating these factors, this study can help shed further light on the role of hepcidin in atherosclerosis and possibly the underlying mechanisms involved.

## 2. Results

### 2.1. Clinical Information

Clinical information from the patients whose samples were studied is given in Appendix A. Atherosclerosis risk factors, such as diabetes mellitus, hypertension, and smoking, were not significantly different among patient groups. In the lipid profile among patient groups, higher levels of high-density lipoprotein (HDL) cholesterol were observed in the asymptomatic patient group as compared with the symptomatic patient group.

### 2.2. Hepcidin Expression Is Significantly Increased with the Progression of Human Atherosclerotic Plaques

In early lesions, hepcidin mainly occurred in the deep intima areas, while in advanced lesions, massive positive hepcidin staining was found and was preferentially expressed in the atherosclerotic intima and in the surrounding regions of necrotic cores. The quantitative analysis showed that hepcidin levels were significantly increased in type 2 (plaques with necrotic cores) and type 3 plaques (ruptured plaques) compared to type 1 lesions (Figure 1A). The levels of hepcidin were further divided into two groups according to the median values of hepcidin and named low or high group (HP low or HP high). As shown in Figure 1B, less than 15% of plaques had higher levels of hepcidin in type 1 lesions, while about 50% were found in type 2 and more than 70% in type 3 plaques.

We further evaluated whether hepcidin expression is related to the clinical symptoms of patients with carotid plaques. We found that the expression of HP, as compared to patients without symptoms (1.24 +/− 0.4), plaques from patients with clinical symptoms expressed relatively higher levels of hepcidin (1.97 +/− 0.3), though not statistically different.

### 2.3. Plaques of Male Patients Have Significantly Higher Levels of Hepcidin

We have previously demonstrated that atheroma of male patients had significantly higher levels of CD68 macrophages, ferritin, and TfR1 in lesions [14]. In the present study, the human carotid plaques were further divided into two groups according to sex. Plaques from male patients showed prominent immunostaining of hepcidin compared with the same type of the plaques from female patients (Figure 2A,B). The quantitative analysis showed that hepcidin levels were significantly higher in the plaques from male patients compared to females (Figure 2C). In contrast to plaques from females, the majority of plaques from male patients had higher levels of hepcidin (Figure 2D). To understand whether the number of hepcidin positive areas between men and women is influenced by the type of plaque present in each gender, we further analyzed the plaque types in relation to the expression of hepcidin in each gender. We found similar results as we demonstrated previously [14] that male patients with carotid atheroma had more advanced and ruptured lesions compared to women. Specifically, women had about 50% type 1, 41% type 2, and 9% type 3 plaques, while men had about 29% type 1, 36% type 2, and 35% type 3 plaques (*p* < 0.05).

### 2.4. Hepcidin Expression Significantly Correlated with the Accumulation of CD68 Positive Macrophages, Transferrin Receptor (TfR1), and Apoptosis

Earlier, we have shown that TfR1 (a major iron importer by internalizing the transferrin-iron complex through receptor-mediated endocytosis) and ferritin (an iron storage and stress protein) are highly expressed in CD68 positive macrophages and were associated with instability and rupture of human carotid plaques [15]. Here, we further examined the expression of hepcidin in relation to CD68 macrophages and iron-related protein ferritin and TfR1. In the serial sections of all studied carotid lesions, the expression of hepcidin was divided into two groups according to median values. The plaques with hepcidin levels ≤ median were defined as the hepcidin low group, while the ones with hepcidin levels > median were defined as the hepcidin high group. The levels of CD68 positive macrophages, SMA (smooth muscle actin), ferritin, TfR1, and TUNEL (terminal deoxynucleotidyl transferase dUTP nick end labeling) were further divided into two groups according to hepcidin levels. As shown in Figure 3, the lesions with higher levels of hepcidin expressed also significantly higher levels of CD68 positive macrophages, while no differences in SMA positive smooth muscle cells. Moreover, we further examined the localization of ferritin, TfR1, and TUNEL in relation to hepcidin on serial sections of carotid lesions. Hepcidin-positive areas corresponded well with TfR1 and TUNEL in the serial sections, while not statistically significant with ferritin though tend to have higher ferritin levels corresponded to higher levels of hepcidin (Figure 3). Significant differences in Hb (hemoglobin) and MCHC (mean corpuscular hemoglobin concentration) were found in patients with higher hepcidin compared with the ones with lower hepcidin (Hb: 145 +/− 2 vs. 137 +/− 2.2, *p* < 0.05; MCHC: 335 +/− 2.4 vs. 317 +/− 11, *p* < 0.05).

### 2.5. Hepcidin Is Significantly Increased in Macrophages Exposed to Iron or 7-Oxysterols (7-Beta-OH or 7-Keto)

Intraplaque hemorrhage and Hb catabolism by macrophages are associated with iron accumulation in the form of ferritin in the progression of atherosclerotic lesions. We reported earlier that both iron and 7-oxysterols can induce ferritin induction in macrophages, which is associated with cellular oxidative stress and apoptosis [16]. However, it is unknown whether iron and 7-oxysterol exposure can induce hepcidin expression in macrophages. To study the effect of cellular iron and 7-oxysterols on hepcidin expression, THP-1 macrophages were exposed to 50 µg/mL FeAC (ferric ammonium citrate) or 28 µM 7beta-OH (7betahydroxycholesterol) or 7keto (7keto-cholesterol), and the expression of hepcidin and ferritin was examined. As seen in Figure 4, iron and 7-oxysterol exposure resulted in a prominent induction of hepcidin, which was predominantly expressed in the cell membranes (Figure 4, upper panel). In parallel with hepcidin expression, iron and 7-oxysterol exposure induced ferritin expression, which was predominantly expressed in the cytosol (Figure 4, lower panel).

### 2.6. Neutralization of Hepcidin Significantly Decreased Cell Death, Oxidative Stress, and LMP in THP-1 Cells Exposed to 7beta-OH

We reported earlier that 7-oxysterol induces macrophage cell death through the induction of cellular ROS (reactive oxygen species) and increased LMP (lysosomal membrane permeabilization). In the present study, we found the correlation of hepcidin with macrophage accumulation and apoptosis, we further asked whether neutralization of hepcidin can protect macrophages from 7-oxysterol-induced cell death and focus on the mechanisms on ROS production and LMP. In this part, the THP1 cells were either treated for 24 h with 7beta-OH, or neutralize hepcidin, or both 7beta-OH and anti-hepcidin antibody. The cells were then stained with acridine orange, or DHE (dihydroethidium) or Annexin V/PI (propidium iodide) for the evaluation of LMP, oxidative stress, or cell death, and analyzed by flow cytometry. As shown in Figure 5, 7beta-OH exposure resulted in LMP, ROS production, and apoptosis, which was significantly inhibited by hepcidin antibody.

## 3. Discussion

Hepcidin, a key regulator in maintaining iron balance and iron recycling, has been found to be increased in circulation in patients with metabolic syndrome and in human atheroma plaques. However, it remains unknown whether hepcidin is related to plaque severity and whether hepcidin expression differs between men and women. In the present study, we demonstrate that the expression of hepcidin is increased with the progression of human atherosclerotic plaque and is associated with macrophage accumulation and apoptosis. The lesions from male patients have significantly higher levels of hepcidin, as well as higher levels of CD68-positive macrophages and TfR1.

Serum hepcidin levels are elevated in metabolic syndrome [8] and are associated with the presence of plaques [9,10,11]. In a hyperlipidemic mouse model, hepcidin deficiency showed a protective effect against atherosclerosis, as it was associated with decreased macrophage iron and a reduced aortic macrophage inflammatory phenotype [17]. Considering the findings mentioned above, our results on the increased levels of hepcidin in advanced lesions in humans, along with macrophage apoptosis, highlight the crucial role of hepcidin in plaque progression and plaque rupture.

Iron accumulation has been found in atherosclerotic lesions, and iron deposition in the plaque, particularly in macrophages, may contribute to plaque development and severity [18]. Hepcidin expression has been found in plaque macrophages of a culprit coronary artery, and circulating hepcidin was elevated in patients with acute myocardial infarction [19]. Moreover, plasma hepcidin was positively associated with mortality of patients with acute coronary syndrome [20] in patients with acute coronary syndrome. It has been demonstrated that hepcidin could be produced by monocytes/macrophages upon inflammation, such as atherosclerosis, to trigger iron retention in macrophages [8]. Consistent with previous findings, we found a significant positive correlation between macrophage accumulation and hepcidin in human carotid plaques. The results indicate that macrophage iron accumulation may aggravate plaque progression and instability.

Sex-specific cardiovascular risk factors are potential influencers of sex differences in atherosclerotic plaque progression and instability. The variation in body iron stores may contribute to the discrepancies in atherosclerosis occurrences between sexes [21]. Furthermore, male gender has been suggested as a predictor for stroke and myocardial infarction [22]. Our research group has previously identified more vulnerable or risky plaques in atherosclerotic lesions from male patients [14]. The present study marks the first instance where we have observed significantly elevated levels of plaque hepcidin in lesions from men compared to women, highlighting a novel finding.

The investigation into hepcidin expression and its association with atherosclerotic plaque instability holds considerable importance for upcoming studies seeking to understand the underlying mechanisms involved. There is a hypothesis that hepcidin might contribute to plaque destabilization by hindering the mobilization of iron from macrophages within atherosclerotic lesions. This could potentially explain the observed lack of increased atherosclerosis in patients with hemochromatosis, a condition characterized by low hepcidin levels [12]. The link between hepcidin expression and atherosclerotic plaque instability is established, primarily attributed to the insufficient clearance of necrotic cells and the activation of matrix-degrading proteases [23]. Hepcidin exacerbates the release of inflammatory cytokines, accumulation of intracellular lipids, oxidative stress, and apoptosis in macrophages with retained iron, further contributing to plaque destabilization [10]. Our previous findings suggest that exposure to 7-oxycholesterol expands the pool of labile iron within cells, inducing ferritin expression, accumulating lipid droplets in the cytosol, and triggering apoptotic cell death in macrophages [16]. In our current study, we observed that 7-oxycholesterol also induces hepcidin expression. Neutralizing hepcidin not only decreased the levels induced by 7-oxycholesterol but also provided protection against cell death by reducing lysosomal membrane permeabilization (LMP) and the generation of reactive oxygen species (ROS). These findings propose a potential new mechanism elucidating the role of hepcidin in atheroma instability and rupture. They also offer insights into a potential therapeutic target for preventing the progression of atheroma and plaque instability.

Limitations of the current study: the primary focus of this study centered on examining the pathology of human carotid atherosclerotic lesions. These lesions were obtained through surgical procedures either due to clinical symptoms or as preventative measures for strokes. Consequently, the findings from this study only provide an indication of potential risks associated with plaque instability and rupture. The sample sizes used in this study were relatively small, resulting in notable variations in tissue expression levels. Therefore, a larger prospective study is necessary to validate our findings.

Our research revealed that neutralizing hepcidin offers protection against cell death induced by 7-oxycholesterol through a reduction in LMP and ROS production. However, further investigation into the signaling mechanisms involved in this model is required. Additionally, it remains unclear whether the 7-oxysterol-induced expression of hepcidin is specific to macrophages, emphasizing the need for further studies involving different types of arterial cells.

In conclusion, hepcidin expression, along with macrophage accumulation and apoptosis with iron retention, may contribute to plaque progression and instability. Sex differences in hepcidin expression, macrophage accumulation, and iron metabolism in atheroma lesions should be considered in clinical diagnosis, progression, and treatment of atherosclerotic diseases. Furthermore, hepcidin plays an important role in macrophage apoptosis induced by oxidized lipids via the lysosomal pathway and oxidative stress.

## 4. Materials and Methods

### 4.1. Collection of Carotid Artery Samples

The atherosclerotic carotid arteries were collected from patients who underwent carotid endarterectomy at Linköping University Hospital. This study was approved by the Regional Ethical Review Board in Linköping (03-499, 2003), and all methods were performed in accordance with the relevant guidelines and regulations. Written informed consent was obtained from all participants.

Carotid samples from 58 patients (38 males and 20 females) were included in the present study. Patients who had not experienced any neurological symptoms within the six months prior to the operation were classified as asymptomatic (Asymp, n = 8), whereas patients with transient ischemic attacks, minor stroke, or amaurosis fugax were considered symptomatic (Symp, n = 50). Several stroke risk factors, including age, hypertension (defined as having a history of hypertension and a diastolic blood pressure ≥ 90 mmHg, all of whom received blood pressure-lowering treatment), smoking (defined as regular smoking for more than five years), and diabetes mellitus (defined as regular administration of diabetes medication), were analyzed. These factors did not show any statistical differences between asymptomatic and symptomatic patients (Appendix A).

Carotid artery samples were collected immediately post-endarterectomy and fixed in 4% formaldehyde. Three to five cross-sectional segments of each specimen were embedded in paraffin.

### 4.2. Immunohistochemistry

Paraffin-embedded carotid arteries were deparaffinized in xylene, rehydrated in graded ethanol (from 100% to 95% and then to 70%), and subjected to immunostaining. Immunohistochemistry was performed on serial sections, as previously described. The primary antibodies used were hepcidin (Santa Cruz Biotechnology Inc., Heidelberg, Germany), CD68 clone PG-M1 (DAKO), smooth muscle actin clone (SMA) 1A4 (DAKO), ferritin (DAKO), and transferrin receptor 1 (TfR1) (Alpha Diagnostic International, TX, USA). The immunoreactions were visualized using the EnVision+/HRP (DAKO) method and ChemMate EnVision Detection Kit (DAKO). Control sections without primary antibodies or with non-immune IgG were run for each protocol, resulting in consistently negative results. The slides were counterstained with hematoxylin.

All histological sections were examined under a light microscope, and the images were digitalized with the Image Grabber program (Toronto, ON, Canada). The microscope was set on the same parameters used to scan all samples. The randomly digitalized images were analyzed with Adobe Photoshop (v5.5), as described previously. The individual responsible for the analysis was blinded to patient information [13].

### 4.3. Terminal Deoxynucleotidyl-Mediated dUTP Nick end Labeling (TUNEL)

To detect apoptotic cells in the tissue sections, the TUNEL method was employed using an in situ cell death detection kit (Roche Molecular Biochemicals, IN, USA) according to manufactory’s instruction, before being visualized with Fast-red.

### 4.4. Classification of the Plaques

To investigate whether the expression of hepcidin was related to plaque progression, all carotid artery samples were classified into three groups based on their morphology and plaque components, as described previously [14]. In brief, the plaques were classified into early and advanced plaques. Early lesions (type 1) were intact plaques without necrotic cores and characterized by the presence of fatty streaks. Advanced lesions were defined as intact plaques (type 2, with an intact fibrous cap, necrotic core formation, and inflammatory cell accumulation) or ruptured plaques (type 3, with a ruptured fibrous cap, often containing a large necrotic core, cholesterol crystals, internal plaque hemorrhage, or thrombosis).

### 4.5. Cell Cultures, Experimental Conditions, and Immunocytochemistry

The THP-1 monocytic cell line was obtained from the American Type Culture Collection (ATCC, Rockville, MD, USA) and cultured in RPMI 1640 medium (Invitrogen, Waltham, CA, USA), supplemented with 10% fetal bovine serum (Invitrogen), and 1% penicillin-streptomycin (Invitrogen). The cells were sub-cultured twice a week and used for experiments. To investigate whether hepcidin is involved in 7oxysterols induced apoptosis, the THP1 cells were either untreated or treated for 24 h with 7betahydroxycholesterol (7beta-OH, 28 μM), or anti-hepcidin antibody (1.25 mg/mL), or both 7beta-OH and anti-hepcidin antibody. The cells were then stained with acridine orange (AO, for lysosomal membrane permeabilization (LMP, 15 min at 37 °C with 2 mL AO solution of 5 μg/mL in complete medium), or dihydroethidium (DHE, Molecular Probes, Eugene, OR, USA) staining for oxidative stress (15 min at 37 °C with 10 µM DHE in cultured medium. For analysis of cell death, the cells were stained with Annexin V/propidium iodide for 20 min on ice (AV/PI, Roche Diagnostics, Mannheim, Germany) and analyzed by flow cytometry.

THP-1 cells were differentiated into macrophages by incubating with phorbol 12-myristate 13-acetate (50 μM for 24 h). Differentiated macrophages were used for experiments after incubation in normal media for two days. In the experiments, the cells were exposed for 24 h either to ferric ammonium citrate (FeAC, 50 μg/mL) or 28 μM 7beta-OH or 7ketocholesterol (7keto) in the culture media. Untreated cells were used as controls. To investigate whether iron or 7beta-OH exposure can induce the expression of hepcidin, immunocytochemistry was performed. Cells were fixed in 4% paraformaldehyde, permeabilized with 0.1% saponin, and incubated with a mouse anti-human hepcidin monoclonal Ab to neutralize hepcidin (1:100, Amegen Inc., Thousand Oaks, CA, USA) or rabbit anti-human ferritin (1:200, DAKO, Carpinteria, CA, USA) at 4 °C overnight. The cells were then incubated with Alexa fluor 488 goat anti-mouse or goat anti-rabbit antibody (Invitrogen, 1:400) for 1 h at room temperature and were mounted with 4′,6-diamidino-2-phenylindole (DAPI)-containing mounting media (Vector Laboratories, Inc., Newark, CA, USA) and analyzed with immunofluorescence microscopy using the 40× oil-immersion objective.

### 4.6. Statistical Analysis

For statistical analyses, a Kruskal–Wallis test followed by Dunn’s post hoc test for multiple comparison was performed, while the Mann–Whitney U test was used for the comparison of 2 groups. Chi-square was used for a comparison of categorical data. Continuous data were presented as mean ± SEM, unless otherwise stated and *p* ≤ 0.05 was considered statistically significant.

## Figures and Tables

**Figure 1 ijms-25-01706-f001:**
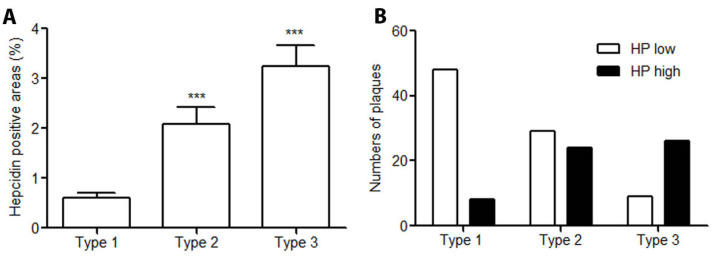
Hepcidin expression was significantly related to atheroma plaque instability. (**A**) Image analyses showed that compared to type 1 (n = 55) lesions, hepcidin expression was significantly increased in both type 2 (n = 53) and type 3 (n = 35) lesions. *** *p* < 0.0001. (**B**) The levels of hepcidin in types 2 (low/high: n = 29/24) and type 3 (low/high: n = 9/26) were predominantly higher than the median level of hepcidin in type 1 (low/high: n = 48/8), while the low level of hepcidin was lesion type dependent, decreasing from type 1 to type 3.

**Figure 2 ijms-25-01706-f002:**
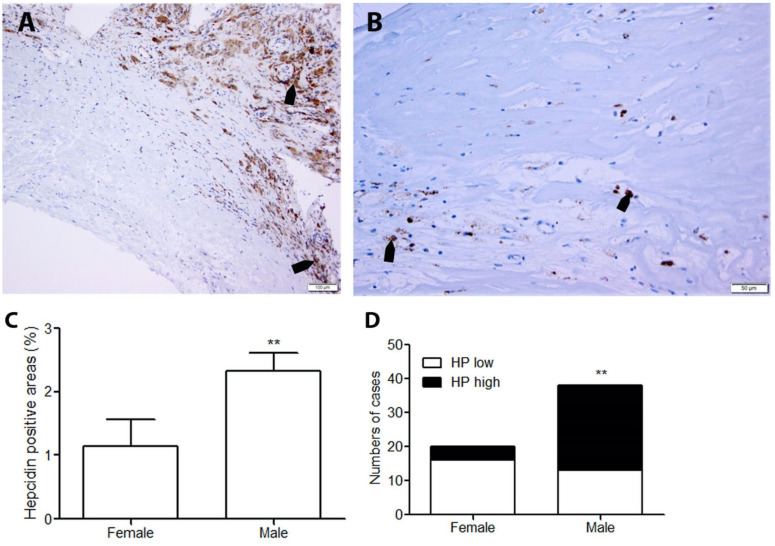
Expression levels of hepcidin were significantly higher in plaques from males than in plaques from females. (**A**,**B**) Representative hepcidin expression in advanced lesions from males (**A**) and females (**B**). Bars = 50 µm. Arrows indicate the positive staining of hepcidin in brown color. (**C**) Quantification of the expression levels of hepcidin in human carotid atherosclerotic lesions by image analysis of immunohistochemistry (n = 38 for men and 20 for women). *p* < 0.01 vs. females. (**D**) The number of male patient cases with high HP level was predominantly higher than the number in female patients. ** *p* < 0.01.

**Figure 3 ijms-25-01706-f003:**
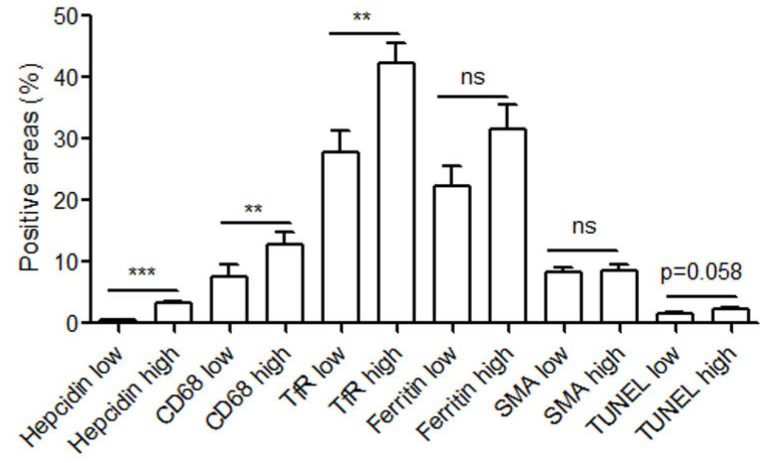
Carotid atherosclerotic lesions with higher levels of hepcidin had significantly higher levels of CD68-positive macrophages and transferrin receptor 1 (TfR1). The serial sections of human carotid plaques were immunostained with antibodies against hepcidin, CD68, smooth muscle actine (SMA), TfR1, or ferritin and stained with terminal deoxynucleotidyl-mediated dUTP nick end labeling (TUNEL). The samples were analyzed by means of image analysis. The patients were divided according to hepcidin expression level (low HP: expression levels lower than median value; high HP: expression levels equal to or higher than median value, n = 29 for both groups). The expression levels of CD68, SMA, TfR1, ferritin, TUNEL were divided according to hepcidin levels (low and high). The bars represent positive areas of each immunostaining in all the samples. ** *p* < 0.01 and *** *p* < 0.001. ns stands for no statistical difference.

**Figure 4 ijms-25-01706-f004:**
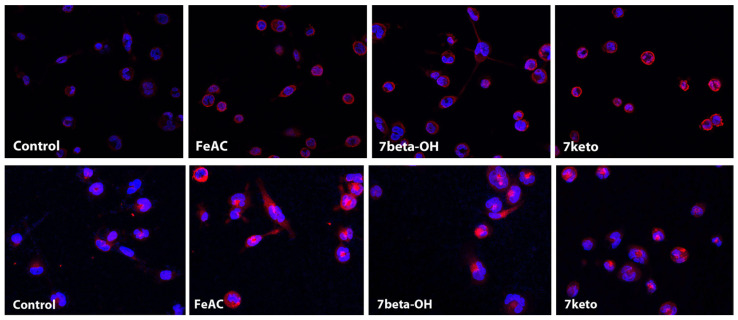
The exposure of iron and 7-oxysterols (7beta-OH and 7keto) causes induction of hepcidin in THP-1 macrophages. THP-1 macrophages were either left untreated or were treated for 24 h with FeAC (50 µg/mL) or 7beta-OH or 7keto (28 µM). The cells were immunostained with hepcidin (upper panel) or ferritin (lower panel) immunocytochemistry and were examined by fluorescence microscopy. Representative photographs of hepcidin or ferritin in red and nuclei in blue with DAPI nuclear staining.

**Figure 5 ijms-25-01706-f005:**
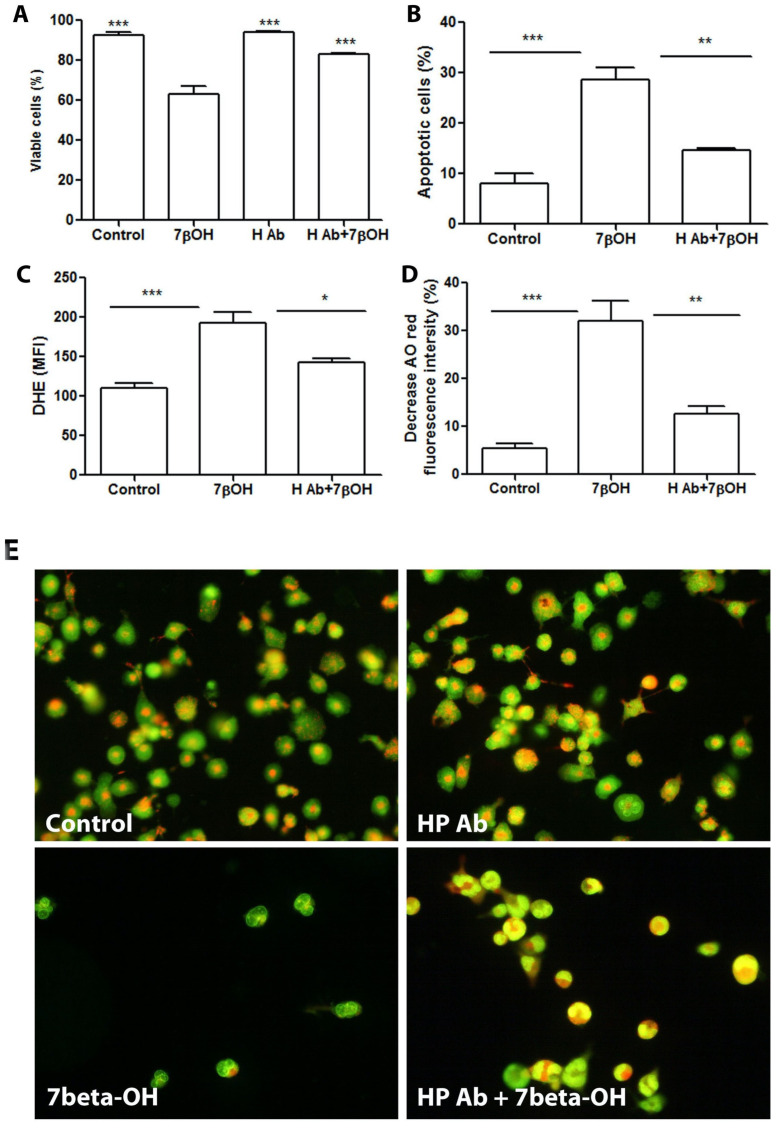
Neutralization of hepcidin significantly decreased cell death, oxidative stress, and LMP in THP-1 cells exposed to 7beta-OH. THP-1 cells were either left untreated or were treated for 24 h with 7beta-OH (28 µM), or with hepcidin antibody (HP Ab), or both. The cells were stained with AVPI, DHE, or AO (acridine orange) and were examined by flow cytometry and fluorescence microscopy. (**A**): Percentage of viable cells. Data presented as mean ± SD, n = 6 for control, 7beta-OH and H Ab + 7beta-OH, n = 4 for H Ab, *** *p* < 0.001 compared to 7beta-OH group. (**B**): Percentage of apoptotic cells. Data presented as mean ± SD. n = 4 for all groups, *** *p* < 0.001 and ** *p* < 0.01. (**C**): DHE immunofluorescence intensity. Data presented as mean ± SD. n = 4 for all groups, *** *p* < 0.001 and * *p* < 0.05. (**D**): Cells with decreased AO red fluorescence intensity. Data presented as mean ± SD. n = 4 for all groups, *** *p* < 0.001 and ** *p* < 0.01. (**E**): Representative photographs of AO-stained cells.

## Data Availability

The data presented in this study are available from the corresponding author on reasonable request.

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
