# Peer review of "Elevated Hepcidin Expression in Human Carotid Atheroma: Sex-Specific Differences and Associations with Plaque Vulnerability"

_ijms, 2024, doi:10.3390/ijms25031706_

Round 1

Reviewer 1 Report

Comments and Suggestions for Authors

I read with great attention and interest the paper entitled "Elevated Hepcidin Expression in Human Carotid Atheroma: Gender-Specific Differences and Associations with Plaque Vulnerability."

I would congratulate the Authors for their initail work

Minor suggestion:

- please, provide a structured abstract

- please, add a specific "paper limitation" paragraph

Reviewer 2 Report

Comments and Suggestions for Authors

IJMS-2803831

The experiments described in this paper have been properly carried out, and have given some interesting results. Also, their description has been clearly done, for the main part, but some aspects of the manuscript require improvement.

In particular, the definition of abbreviations is unfortunate and affects readability of the paper. IJMS’s ‘Instructions for Authors’ is clear:

Acronyms/Abbreviations/Initialisms should be defined the first time they appear in each of three sections: the abstract; the main text; the first figure or table.

The authors do not follow this instruction. Definition of several abbreviations (FeAC, SMC, TUNEL, LMP) is deferred to section 3.Discussion or 4.Materials and Methods. Other abbreviations (chemicals: AVPI, DHE, AO) are not defined at all, although they are not standard chemicals (such as ATP, EDTA). You may think that every medical researcher should know what MCHC means, but if you leave out its definition, your paper becomes less readable for potentially interested readers from other fields, e.g., biochemists. Also, in Fig 3 the graphics claim measurement of SMC, while the legend claims measurement of expression levels of SMA (which is nowhere defined).

A statistical correlation between two observables is not equal to a mechanism, not by a long shot. In the Introduction section, the authors announce that they will investigate ‘the underlying mechanisms’.  I failed to find the description of any mechanism anywhere in the paper. In the Discussion section the authors claim that ‘Our findings suggest a new mechanism behind the role of hepcidin in atheroma instability and rupture’. Please, if you think that you have measured, or discovered, or even only thought of a mechanism, then spell it out, so that future researchers will have something to shoot at.

Minor points:

Line 19:                7beta-hydrocycholesterol  à  7beta-hydroxycholesterol

Line 35:                Dr. Sullivan  à  Sullivan

Line66:                 7beta-cholesterol  à  7beta-hydroxycholesterol  (also Line 291)

Line 114:             iron importer  à  iron-transferrin importer

Line 252:             graded ethanol  à  What grade?

Line 291:             50 ug/mL  à  190 uM

Reviewer 3 Report

Comments and Suggestions for Authors

The authors present an interesting study in which the influence of hepcidin in plaque severity is examined. Briefly, the authors employ clinical samples to first determine the association with hepcidin and plaque presence across a biobank of samples covering a number of plaque types and also gender. Based on these samples, hepcidin presence correlated with plaque severity, and the authors then pivot to in vitro studies to explore whether therapeutically targeting hepcidin and iron turnover reduced clinical risks. Based on the macrophage response, suppression of hepcidin reduced risk to macrophage cells suggesting clinical targeting of such may reduce plaque severity.

In reviewing the manuscript, I made a number of observations. The following should be considered by the authors when preparing a suitable revision.    

1.       Table 1 was missing from the document I reviewed, and thus many questions were formulated as a result of this information being absent. This table needs to be included in the article such that the information provide can be reviewed in the context of the study.

2.       In Figure 2, arrows should be included to indicate the hepcidin positive areas in the images

3.       When comparing the number of hepcidin positive areas between men and women, is this influenced by the type of plaque present in each gender? Was any analysis done to clarify whether males are a t higher risk of a particular type of plaque as compared to females, and does this influence the amount of hepcidin positive areas as a result?

4.       In Figure 3, more information is required to determine what this data represents. Do the bars represent all off the samples, or a particular subset?

5.       The methods require more detail. There are details missing such as the source of the reagents/chemicals, the concentrations used (e.g. antibodies), and important steps (e.g. cell culture, microscopy, etc.). These aspects and others require more attention in the interest of a reader potentially having the information needed to repeat the experiment.

6.       N-numbers should be clearly stated in each figure.

Round 2

Reviewer 3 Report

Comments and Suggestions for Authors

The authors have suitably addressed my comments.